Dental microwear reveals mammal-like chewing in the neoceratopsian dinosaur Leptoceratops gracilis

Varriale Frank J. frankvarriale@kings.edu
Department of Biology, King’s College , Wilkes-Barre , PA , United States
Wedel Mathew
Electronic publication date: 2016 Jul 6
Publication date: 2016
Volume: 4
Electronic Location ID: e2132
Received 2016 Mar 23; Accepted 2016 May 24
Copyright: ©2016 Varriale
Copyright year: 2016
Copyright holder: Varriale
License: This is an open access article distributed under the terms of the Creative Commons Attribution License, which permits unrestricted use, distribution, reproduction and adaptation in any medium and for any purpose provided that it is properly attributed. For attribution, the original author(s), title, publication source (PeerJ) and either DOI or URL of the article must be cited.
License URL: https://creativecommons.org/licenses/by/4.0/

Keywords: Dental microwear, Jaw action, Mastication, Chewing, Ornithischia, Dinosauria, Ceratopsia, Jaw mechanics

Funding: Jurassic Foundation Sigma Xi The Geological Society of America Stephen J. Gould Award from the Paleontological Society Funding for this research was provided by grants from the Jurassic Foundation, Sigma Xi: Grants in Aid, The Geological Society of America, and a Stephen J. Gould Award from the Paleontological Society. The funders had no role in study design, data collection and analysis, decision to publish, or preparation of the manuscript.

==============================
Extensive oral processing of food through dental occlusion and orbital mandibular movement is often cited as a uniquely mammalian trait that contributed to their evolutionary success. Save for mandibular translation, these adaptations are not seen in extant archosaurs or lepidosaurs. In contrast, some ornithischian dinosaurs show evidence of precise dental occlusion, habitual intraoral trituration and complex jaw motion. To date, however, a robust understanding of the diversity of jaw mechanics within non-avian dinosaurs, and its comparison with other vertebrates, remains unrealized. Large dental batteries, well-developed dental wear facets, and robust jaws suggests that neoceratopsian (horned) dinosaurs were capable chewers. But, biomechanical analyses have assumed a relatively simple, scissor-like (orthal) jaw mechanism for these animals. New analyses of dental microwear, presented here, show curvilinear striations on the teeth of Leptoceratops. These features indicate a rostral to caudal orbital motion of the mandible during chewing. A rostrocaudal mandibular orbit is seen in multituberculates, haramiyid allotherians, and some rodents, and its identification in Leptoceratops gracilis is the first evidence of complex, mammal-like chewing in a ceratopsian dinosaur. The term circumpalinal is here proposed to distinguish this new style of chewing from other models of ceratopsian mastication that also involve a palinal component. This previously unrecognized complexity in dinosaurian jaw mechanics indicates that some neoceratopsian dinosaurs achieved a mammalian level of masticatory efficiency through novel adaptive solutions.

Introduction

Mastication, or chewing, is the process by which food items are broken down between precisely occluding teeth via repeated mandibular adduction (Hiiemae, 2000). Often assumed as an exclusively mammalian feature, the process of mastication increases the surface area of ingested food for subsequent enzymatic action in the digestive system, allowing more efficient energy extraction. However, mounting evidence indicates that the distribution of chewing is more widespread in non-mammalian synapsids (Crompton & Hotton, 1967; Crompton, 1972; Crompton & Attridge, 1986; King, Oelofsen & Rubidge, 1989; King, 1990; King, 1996; Cox, 1998; Angielczyk, 2004) and sauropsid amniotes than previously thought (Weishampel, 1984; Norman & Weishampel, 1985; Rybczynski & Vickaryous, 2001; Ösi & Weishampel, 2009; O’connor et al., 2010). Among non-avian dinosaurs, mastication is well supported in hadrosaurid ornithopods. Hadrosaurs possess numerous cranio-dental specializations for mastication, including a jaw joint depressed below the dental arcade, an elongated coronoid process, and most strikingly a dental battery consisting of many closely packed teeth with dentine of differing hardness (Ostrom, 1961; Weishampel, 1984; Norman & Weishampel, 1985; Williams, Barrett & Purnell, 2009; Erickson et al., 2012; Cuthbertson et al., 2012). Many neoceratopsians exhibit similar adaptations, providing support for an identical level of masticatory sophistication (Ostrom, 1964; Ostrom, 1966; Tanoue et al., 2009; Erickson et al., 2015). Because neoceratopsian skulls lack intracranial joints with large gaps (Holliday & Witmer, 2008), and have a simple hinge-like jaw mechanism, biomechanical analyses of their chewing infer an unsophisticated orthal motion that resulted in a scissor-like (orthal) adduction of the lower jaw (Ostrom, 1964; Ostrom, 1966; Tanoue et al., 2009).

Food often leaves microscopic traces known as microwear, in the form of pits and scratches on the occlusal surfaces of teeth (Teaford, 1988a; Teaford, 1991). Comparison of microwear within and among taxa and analyses of scratch orientation have been successfully used to test hypotheses of jaw motion in numerous mammalian taxa (Teaford & Byrd, 1989; Teaford, 1991; Charles et al., 2007), as well as various dinosaur groups (Weishampel, 1984; Upchurch & Barrett, 2000; Rybczynski & Vickaryous, 2001; Williams, Barrett & Purnell, 2009; Whitlock, 2011; Mallon & Anderson, 2014; Ösi et al., 2014). Unlike mammalian teeth, each neoceratopsian tooth possessed a single, relatively flat, near vertical occlusal surface resulting from shear between the maxillary and mandibular dentitions (Hatcher et al., 1907; Dodson, Forster & Sampson, 2004). Occlusal microwear on ceratopsian teeth is the result of the direction and magnitude of the jaw closing power stroke of mastication, and an exceptional record of jaw action is recorded on teeth as a result of the planar nature of the occlusal surface. Given a strictly orthal model, predicted microwear should be composed of striations that are rectilinear and limited to a single modality at or near the apicobasal axis of the tooth. To test this hypothesis, dental microwear was examined in the non-ceratopsid neoceratopsian Leptoceratops gracilis, B. Brown (Brown, 1914).

Materials & Methods

Dental microwear was examined in a nearly complete, articulated skull of the neoceratopsian Leptoceratops gracilis (CMN 8889; Canadian Museum of Nature, Ottawa, Ontario, Canada) as well as the holotype material (AMNH FR 5205; American Museum of Natural History, New York, USA) and isolated teeth (n = 2) assigned to the taxon (YPM VPPU 018133; Yale Peabody Museum of Natural History, New Haven, Connecticut, USA). The conclusions in this analysis are derived primarily from examination of CMN 8889. AMNH FR 5205 and YPM VPPU 018133 were examined to provide qualitative support for conclusions drawn from CMN 8889; they were not quantitatively analysed because preserved microwear was either of slightly lesser quality and did not provide a large sample size (AMNH FR 5205) or represented isolated teeth (YPM VPPU 018133).

Each tooth was molded using Coltene President surface activated polyvinylsiloxane molding compound. This compound is available in multiple thicknesses, and the President Jet regular (product number C6012) was used to obtain peels of occlusal surfaces, whereas the President impression putty (product number C4843) was used to create walls around the edge of peels, transforming them into cups capable of receiving epoxy resin. Prior to molding, teeth were screened for alterations and artifacts that can occur due to taphonomic, post depositional, and museum conservation processes. Teeth from all specimens included here were deemed suitable for analysis because they met a number of criteria suggested by other workers as indicative of genuine microwear. Microwear on the Leptoceratops sample is relatively uniform, having a regular pattern and showing no evidence of multiple abrupt shifts or large gouges indicative of preparation marks (Teaford, 1988b). Striations are well defined and show none of the obliteration and dulling characteristic of taphonomic particulate abrasion or acid etching (Teaford, 1988b; King, Andrews & Boz, 1999). When preservative obscuring occlusal surfaces was encountered, it was removed using a gentle scrubbing action with cotton swabs and acetone or ethanol as the solvent.

Casts were poured using a two-part epoxy resin (EPO-TECH #301) designed to cure slowly at room temperature with minimal exothermal heat production. Casts were allowed to cure for 14–21 days at ≈23°C. Upon solidification, casts were removed from their molds and secured to SEM specimen mounts (SPI Supplies® Aluminum Pin-Type Mounts #1507L-MB) using a nonconductive modified nitrocellulose solution (Duco Cement®). Casts were then gold-palladium coated for 180 s using a Denton Vacuum Desk III set to 40 milliamps and a 50% Argon gas mixture. Silver paint (SPI Supplies®, Ag Colloidal Suspension #05001-AB) was used to create a conductive connection between the coated specimen and SEM mount, ensuring electron transmission.

Microscopic examination was conducted using an Amray 1810 scanning electron microscope set to a working distance of 11 mm, and a 20-keV electron beam in secondary emission mode. The occlusal surface was oriented orthogonal to the electron beam to provide a controlled position. This, coupled with the secondary emission mode, helped to minimize the extinction of features that can occur with tilting and use of backscatter electrons (Galbany, Martinez & Perez-Perez, 2004). Specimens were photographed at 100X using Polaroid®, Polapan 55 film (ISO 50/18° 20 s/Sek/s), and micrographs were scanned as bit-map images at a resolution of 300 dpi for computer analysis.

Wear features were digitized using Microware, Version 4.02, a semi-automated image analysis system for the quantification of dental microwear (Ungar, 2002). Microware 4.02 records features as four points (x, y coordinates) on a Cartesian grid as the user defines the length and width of a feature by clicking on its ends with a computer mouse. These Cartesian data were imported into a Microsoft Excel spreadsheet, and a macro (Articles S1 and S2) was written to trigonometrically calculate the angle of microwear striations within a 180° arc progressing from apical to basal on the distal side of the tooth.

Individual teeth were examined separately, because whole jaws cannot fit into the SEM chamber. This necessitated directional standardization within the chamber so that micrographs and angular data would be consistent and comparable among teeth. The primary ridge of the tooth crown is a prominent feature common to all teeth, and its orientation was assumed to be near apicobasal. The apex of the tooth was oriented to the left using this ridge and micrographs were then photographed in this position. Some micrographs were flipped horizontally using Adobe Photoshop prior to digitization with Microware 4.02 to obtain angular data directly comparable and easily visualized for all teeth in all quadrants of the dentition. Micrographs from teeth in the right dentary quadrant were designated as the standard and all other quadrants manipulated to provide data comparable with this quadrant. Left dentary and right maxillary micrographs were flipped horizontally. Left maxillary teeth did not require manipulation because angles measured from this quadrant are the same as those in the right dentary. This protocol produced comparable angles without the need for mathematical transformation via the trigonometric principle that alternate interior angles are equal. Analyses of angular data (Data S1) were conducted using Oriana, Version 2.0 (Kovach, 2003), a circular statistics and rose diagram software package. Oriana was used to generate rose diagrams, calculate mean angle, Rao’s spacing test, and length of the mean vector (r). Rose diagrams depict angular data on a unit circle with increasing angle as a function of clockwise rotation. As such, all rose diagrams herein depict angles relative to wear on a left dentary tooth of a left facing skull, facilitating direct visual comparison among dental quadrants. In all rose diagrams 90° is equivalent to the caudal direction and 180° is ventral.

r and mean angle

The homogeneity of scratch orientations can be measured by the value r, which is the length of the mean vector of circularly-distributed data on a unit circle with an imaginary radius of 1.0 (Zar, 1998; Mardia & Jupp, 2000). The mean vector is inversely proportional to uniformity, with values ranging from 0 to 1.0. Values of r approaching 0 indicate striations whose angles are uniformly dispersed around the circle, whereas those approaching 1.0 indicate angles with high homogeneity confined to a relatively small angular arc. The following formula describes r: r=1n∑i=1n cosθi2+∑i=1n sinθi2

where n is the sample size and θ is the angle of the ith striation. Because a single axis can be described by two different angles 180° apart, using these data in the above equation without transformation would yield an r value lower than expected. For example, two striations with angles 10° and 190° are parallel to each other and striking in the same direction, thus an r value of 1.0 is expected. However, these values yield an r = 0 when inserted into the equation because they are on opposite sides of the unit circle and thus evenly distributed. For axial data, all angles are doubled (multiplied by 2), and any angle greater than or equal to 360° is subtracted by 360° . This transformation has the effect of rotating any angle of θ that is in the 180°–359° hemisphere back into the 0°–179° hemisphere so that it is equal to its counterparts in that half of the circle, and yielding a correct r value. The mean angle can then be found using the mean sine and cosine values calculated above for the length of the mean vector, using the inverse tangent function: θ¯=tan−1∑i=1n sinθi∑i=1n cosθi.

Because of the doubling of angles for the r calculation above, the angle resulting from this equation should be halved to arrive at the correct mean angle (Zar, 1998; Mardia & Jupp, 2000).

Rao’s spacing L-test

Rao’s spacing test evaluates the null hypothesis that a sample of angles is uniformly distributed around a circle (Mardia & Jupp, 2000). A value less than the predetermined significance level rejects the null hypothesis and supports the alternative, that the angles show a preferred direction. This test was chosen because it is more powerful when confronted with bimodal data separated by 180° , compared to other similar tests such as the Rayleigh test of uniformity. The Rayleigh test incorporates r in calculating its test statistic Z(Z = nr2). Because bimodal data on opposite sides of the circle yield low r values, the Z value would be proportionately small. Small Z values fail to reach the given level of significance (p = 0.05), increasing the likelihood of a type II error, which would fail to reject the null hypothesis. Rao’s spacing does not use r, but instead examines the spacing between points on the unit circle and compares their deviation from the uniform case where the spacing should be 2π∕n radians (360° /n). The L statistic is calculated: L=12∑i=1nTi−2πn

where n is the sample size, and Ti is the spacing or difference between the observed angles given the following: Ti=θi−θi−1,i=1,…,n−1,Tn=2π−θn−θ1.

Because Ti is the distance between n observed points and 2π∕n is the expected distance between n points then the difference between them should be large if the observed points are clustered, thus yielding a large value of L (Mardia & Jupp, 2000).

Results

Microwear striations on teeth of Leptoceratops (CMN 8889) conform to predictions of orthal chewing by being unimodal in distribution (Fig. 1). However, they differ greatly in other parameters. Striations are not rectilinear, as would be expected of a simple scissor-like closure of the mandible, but are curvilinear (Figs. 2A–2B and 3). Striations begin at the apicodistal border on dentary teeth and curve though an arc of nearly 50° before ending in either a horizontal or apicomesial orientation near the mesial border (Figs. 2A and 3A–3B). Maxillary teeth show a counterpart curvature, with striations beginning apicomesially and ending distal to apicodistally (Figs. 2B and 3C–3D).

Figure 1 Histogram of striation orientations pooled from all teeth examined in Leptoceratops gracilis (CMN 8889).

Classes are in 10° increments. A single mode is present, with the greatest frequency of scratches in an arc from 30° to 60° . Line intersecting largest bar is the pooled sample mean angle (44.1°). N = 1,504.

Figure 2 Dental microwear on representative teeth of Leptoceratops gracilis (CMN 8889).

(A) Light microscope images of the seventh left maxillary, and (B) sixth left dentary teeth showing curvilinear microwear traversing the entire occlusal surface. Arrows indicate the direction of initiation and exit of the power stroke. Microwear near the site of initiation is oriented caudodorsally whereas the same wear near the end of the stroke is oriented rostrocaudally, a ≈50° shift in orientation. Dotted lines in (A) and (C) demarcate approximate junction of hard mantle dentine (HMD) with orthodentine (O). (C, D) SEM micrographs imaged from rectangles indicated in (A) and (B) respectively. At this magnification (100×) wear appears uniform with a striation dominated texture. (E, F) Rose diagrams of angular data from striations in (C) and (D). Rose diagrams are on a unit circle and summarize angular orientation relative to wear on a left dentary tooth. 90° = caudal and 180° = ventral directions. Values in lower right quadrant of rose diagrams are the mean angle and the length of the mean vector (r). Arrows in rose diagrams indicates direction and magnitude of r.

Figure 3 Light microscope images of additional teeth from Leptoceratops gracilis (CMN 8889) showing semicircular dental microwear.

(A) Eighth right dentary tooth. (B) 12th right dentary tooth. (C) 10th right maxillary tooth. (D) 11th right maxillary tooth. Colored arrows correspond to those in Fig. 6 and indicate (where visible) the overall orientation of microwear at the initiation (blue) and ending (red) of the power stroke. Arrow (black) in (D) indicates step between two facets (f1, f2) formed via differential wear. These facets were likely caused by the occlusion of this tooth against two opposing right dentary teeth at different stages of eruption. Images not to scale.

As a result of this curvature, angular measurements from multiple locations on the occlusal surface yield different values and are thus not directly comparable. To foster comparability, measurements were taken from micrographs imaged near the apicodistal limit of dentary teeth and the apicomesial edge of maxillary teeth, where the beginning phase of the power stroke initiates microwear formation. The occlusal surface of Leptoceratops teeth is composed of the enamel, hard mantle dentine, and orthodentine materials discussed by Erickson and others (2015). Microwear is formed in, and traverses all of these tissues, but only a thin rim of enamel encapsulates the edge (cutting edge) of the occlusal surface, As such, most of the occlusal surface is hard mantle dentine and orthodentine. As mentioned above, micrographs were taken near the apicodistal and apicomesial edges of teeth, resulting in capture of microwear formed primarily in orthodentine and some hard mantle dentine. Microwear can be seen to traverse materials of differing hardness without appreciable shift in direction (Figs. 2A and 2C).

A Rao’s spacing test was performed on angular measurements from each tooth (Table 1), and in all cases a uniform distribution of measurements about a 360° circle is rejected (p < 0.01), supporting a preference for the mean direction. Furthermore, striations are not oriented with or near the apicobasal axis of the tooth but are inclined distally relative to it. This distal inclination is consistent across all teeth examined with means from individual teeth ranging from 26°–55° (Table 1 and Figs. 2C–2F and 4).

Table 1 Summary and inferential statistics for angular data from teeth examined in Leptoceratops gracilis (CMN 8889).

Alveolar position of each tooth from rostral is indicated by number.

Maxillary teeth	LM5*	LM6	LM7	LM13*	RM2*	RM3*	RM4	RM10	RM11	RM13*	
N	142	100	130	104	37	71	84	103	39	8	
Mean θ∘	43.56	47.15	26.85	39.63	31.23	38.66	40.63	35.3	34.87	46.6	
Length of θ∘(r)	0.686	0.646	0.533	0.833	0.98	0.867	0.897	0.665	0.9	0.993	
Rao’s spacing (U)	198.208	203	191.123	246.954	308.541	243.248	264.029	216.012	260.246	292	
Rao’s spacing (p)	<0.01	<0.01	<0.01	<0.01	<0.01	<0.01	<0.01	<0.01	<0.01	<0.01	
Dentary teeth	LD4*	LD6	LD7*	LD13	RD8	RD9*	RD10*	RD12	RD14*	
N	30	129	67	175	61	59	58	76	31	
Mean θ∘	53.65	54.44	55.09	43.44	53.82	51.29	52.96	43.61	40.78	
Length of θ∘(r)	0.905	0.884	0.886	0.548	0.85	0.976	0.987	0.9	0.958	
Rao’s spacing (U)	256	259.712	252.642	213.686	288.39	301.197	311.779	267.116	280.574	
Rao’s spacing (p)	<0.01	<0.01	<0.01	<0.01	<0.01	<0.01	<0.01	<0.01	<0.01	
Notes.

* Data for additional teeth not figured in text.

LD left dentary

LM left maxillary

RD right dentary

RM right maxillary

Figure 4 Micrographs of selected teeth from each of the four dental quadrants in Leptoceratops gracilis (CMN 8889).

(A) 13th left dentary tooth, (B) sixth left maxillary tooth, (C) eighth right dentary tooth, and (D) fourth right maxillary tooth. Orientations of micrographs are as follows; apical at left and basal at right. In left dentary and left maxillary micrographs distal is at top and mesial is at bottom. In right dentary and right maxillary images distal at bottom and mesial is at top. Rose diagrams summarize angular orientation relative to wear on a left dentary tooth. Description of rose diagram orientation and data are the same as in Fig. 2.

Many striations traverse the entire occlusal surface unbroken, indicating that each was caused by particulates being dragged continuously across the tooth surface (Figs. 2A–2B and 3). Striations also show a high length of the mean vector (r), ranging from 0.53 to 0.99 for individual teeth (Table 1 and Figs. 2E–2F and 4). Length of the mean vector is a statistic that summarizes the parallelism of scratches. If all scratches are nearly parallel, or have a similar angular orientation, then the value of r approaches one, if they are more randomly distributed then this value approaches zero (Zar, 1998; Mardia & Jupp, 2000).

Microwear preserved on additional specimens of Leptoceratops (AMNH FR 5205, YPM VPPU 018133) corroborates qualitative observations made on CMN 8889. An isolated right maxillary tooth assigned to Leptoceratops (YPM VPPU 018133) shows dental microwear with an apicomesial (rostroventral) orientation near the site of powerstroke initiation, and striations curve through a similar arc to end in a mesiodistal (rostrocaudal) orientation (Compare Fig. 5A with Fig. 2A). Teeth from AMNH FR 5205 show multiple stages of wear. On some teeth, microwear is interrupted by preparation marks, broken crowns, or what appear to be root erosions. Some microwear is visible on the occlusal surface of the fifth right dentary tooth (Fig. 5B), despite missing the apex and mesial half of the crown. The dominant wear grain consists of striations oriented parallel to subparallel with the mesiodistal axis, and showing a slight change in curvature. The wear also reflects a similar orientation as the nearby labial shelf (LS). Considering this structure as a result of differential wear via incomplete shear of maxillary teeth against their dentary counterparts, its orientation should be similar to nearby microwear on the occlusal surface.

Figure 5 Light microscope images of dental microwear in additional specimens of Leptoceratops.

(A) Isolated right maxillary tooth assigned to Leptoceratops (YPM VPPU 018133) showing semicircular dental microwear. (B) Fifth right dentary tooth from Leptoceratops gracilis (AMNH FR 5205) showing a predominance of mesiodistally oriented wear near the base of the facet, and in a similar orientation as the labial shelf (LS). Colored arrows correspond to those in Fig. 6 and indicate the overall orientation of microwear at the initiation (blue) and ending (red) of the power stroke. Images not to scale.

Figure 6 Model of circumpalinal mastication in Leptoceratops.

(A) Adduction of the lower jaw dominated by action (blue arrow) of the m. adductor mandibulae externus group (mAME) and beginning of the power stroke of mastication. (B) Progression of the power stroke into a palinal (retraction) phase dominated by transition from the mAME to action (red arrow) of the m. addcutor mandibulae posterior (mAMP). Below each skull is a left dentary tooth with colored arrows demarcating the segment and direction of microwear striations that result from the aforementioned muscular actions.

Discussion & Conclusions

The inclination and curvilinear character of microwear striations is entirely inconsistent with the standard orthal model (Ostrom, 1964; Tanoue et al., 2009) of neoceratopsian mastication. A significant Rao’s spacing test supports a power stroke that was directed along the mean striation angle. To traverse an unbroken arc, the mandible must have undergone uninterrupted precise motion as the dentary teeth slid past the maxillary teeth during the power stroke. Indeed, the high degree of striation parallelism (high r) indicates a power stroke event that was performed under precise muscular action and one that must have been stereotyped due to the homogenous nature of curvilinear striations. (Figs. 2–5).

Refutation of the orthal model of mastication in neoceratopsians requires an alternative with increased explanatory power. Unlike mammals, ceratopsians retained the plesiomorphic organization of jaw adductor muscles present in their amniote ancestors. The m. adductor mandibulae externus group and the m. adductor mandibulae posterior are the major jaw closing muscles (Haas, 1955; Holliday, 2009). These muscles have caudodorsal vectors, with the former having a stronger dorsal component and the latter a more caudal orientation. The m. pterygoideus group is also involved in jaw closure, having a rostrodorsal vector. However, its role is more pronounced during the beginning of adduction (Carroll, 1969) rather than the power stroke, and a correlate in microwear is undetectable. Microwear at the apicodistal limit of dentary teeth is oriented in the direction of the supratemporal fenestrae, suggesting that the power stroke was initiated by the m. adductor mandibulae externus group. Although the m. adductor mandibulae externus group would have assisted in retraction of the jaw due to its caudodorsal vector, the muscle of primacy for producing the mesiodistally oriented microwear at the basomesial edge of dentary teeth was the m. adductor mandibulae posterior. Production of the uniform arc seen on these teeth must have involved a precise yet smooth transition between actions of the m. adductor mandibulae externus group and the m. adductor mandibulae posterior (Fig. 6).

Microwear in Leptoceratops supports a model of mastication where the initiation of the power stroke was simultaneously orthal and palinal. Propalinal motion has been proposed in various ornithopods, and the sauropod Diplodocus (Ostrom, 1961; Barrett & Upchurch, 1994; Williams, Barrett & Purnell, 2009; Williams, 2010). Pilot work by Sampson (1993) and Barrett (1998) suggested propalinal motion was present in the derived ceratopsid, Triceratops. However, the mechanism proposed here for Leptoceratops is of a decidedly palinal nature, with no forward motion of the mandible while the teeth are in occlusion. Mallon & Anderson (2014) examined a variety of Campanian hadrosaurids and did not recover the multiple scratch classes indicative of propaliny found by Williams and others (2009), instead, finding only a bimodal distribution with classes in the dorsoventral to caudodorsal direction. They conclude that an orthopalinal powerstroke characterized hadrosaurid mastication. An orthopalinal direction was likewise proposed for Ceratopsidae by Varriale (2011), and his work has been upheld by Mallon & Anderson (2014) who also examined ceratopsid microwear. Whereas the initiation of the powerstroke in Leptoceratops agrees with the orthopalinal models of Varriale (2011) and Mallon & Anderson (2014), the full cycle of the stroke departs strongly. This initial inclined palinal direction has also been demonstrated in the basal ceratopsian Psittacosaurus. Previously, Sereno (1987) and Norman & Weishampel (1991) suggested a propalinal mechanism for Psittacosaurus, with Sereno (1987) forwarding a palinal powerstroke. However, recent work has clarified a more detailed mechanism where inclined dental facets combined with diverging tooth rows and an orthopalinal motion yielded continuous occlusion during the power stroke (Sereno, Xijin & Lin, 2010). The term clinolineal was coined to encompass the jaw mechanism of Psittacosaurus, because occlusion occurred over an inclined linear direction (Sereno, Xijin & Lin, 2010). However, microwear produced by clinolineal and orthopalinal chewing is rectilinear and not strongly curved, as that demonstrated here in Leptoceratops. Unlike the condition in Psittacosaurus and derived ceratopsids, the semicircular microwear of Leptoceratops supports a power stroke that progressed smoothly into a palinal phase (Fig. 6). The term circumpalinal is proposed here to describe the semicircular orbit that is accomplished during the overall palinal (front-to-back) jaw action in Leptoceratops, and to distinguish this style of chewing from the clinolineal and orthopalinal mastication of other ceratopsians.

Among dinosaurs, complex mastication may have been achieved through intracranial joints, in a manner very different from that seen in mammals. Euhadrosaurs (duck-billed dinosaurs) have been reconstructed as masticating using pleurokinesis, a unique motion in which the maxillae and associated teeth swung laterally as the mandibular dentition occluded with them (Weishampel, 1984; Norman & Weishampel, 1985). However, the existence of pleurokinesis is currently contested. The work of Holliday & Witmer (2008) indicates that despite having intracranial synovial joints, hadrosaurids were only partially kinetically competent. Cuthbertson and others (2012) further examined the masticatory apparatus in Brachylophosaurus and Edmontosaurus by scrutinizing dental microwear, arthrology, and kinematic models. They concluded that the facial skeleton of these taxa were akinetic. Still, microwear from the teeth of euhadrosaurs indicates that the jaws may have slid past one another in a pleurokinetic mechanism (Williams, Barrett & Purnell, 2009). However, it now seems that mandibular long-axis rotation, combined with an orthopalinal powerstroke and accessory propalinal motion, has the greatest explanatory power for interpreting observed microwear (Cuthbertson et al., 2012; Nabavizadeh, 2014; Mallon & Anderson, 2014).

Depending on the phylogenetic position of Heterodontosaurus, basal ornithischians may also have chewed by rotation of the dentary bones about their long axes (Hopson, 1980; Weishampel, 1984), producing an effect similar to pleurokinesis. Crompton & Attridge (1986) rejected mandibular rotation based on the presence of planar wear facets and a mediolaterally expanded jaw joint. They substituted a mechanism of medial deviation of the mandibular rami by decreasing the lingual angle at the mandibular symphysis. Recently, Norman et al. (2011) contested both mandibular rotation and medial flexion, citing several restrictions that would have prevented these motions, but they did not provide an alternative mechanism. Sereno (2012) questioned some of the restrictions enumerated by Norman et al. (2011), such that the remaining details and further analysis by him supported mandibular rotation with some medial flexion.

What little is known of their mastication indicates that each of the major ankylosaur clades may have had their own unique and complex jaw actions. Rybczynski & Vickaryous (2001) showed that mastication in the derived ankylosaurid Euoplocephalus was accomplished by swinging the caudal portion of the dentary bones laterally as a result of translation within the jaw joint. The predentary and dentary bones forming a condyloid joint, permitting medial rotation of the anterior dentary. The combined effect created a propalinal arcing of the mandible with both shearing and crushing abilities (Rybczynski & Vickaryous, 2001). Ősi and others (2014) examined microwear, jaw architecture, and myology in the nodosaurid Hungarosaurus, reconstructing a different action than that in Euoplocephalus. Microwear on dentary teeth of Hungarosaurus is bimodal with a near apicobasally oriented class that traverses most of the occlusal surface, and a mesiodistal class that is principally located near the base of occlusal facets. They propose that jaw action was initiated with an orthal motion followed by transition into a palinal powerstroke, these actions being assisted by mandibular rotation or medial flexion.

The action seen here in Leptoceratops bears some resemblance to that proposed for Hungarosaurus (Ősi et al., 2014); however, there are also some notable departures. The apicobasally/apicodistally oriented microwear on Hungarosaurus dentary teeth does not show a consistent and continuous transition to a mesiodistal orientation. Save for a few scratches showing a curved transition, the two classes are disconnected, indicating that transition to a palinal stroke was not as smooth as in Leptoceratops, but a discrete event. The proposed chewing cycle for Hungarosaurus bears this out, with orthal and palinal phases depicted as discreet and rapid changes in vector (Ősi et al., 2014). The transition in Leptoceratops was not swift and abrupt, but one spread over the entire powerstroke with the possibility of one muscle group trading action to another. The relatively flat glenoid of Hungarosaurus compared to the deeply cupped joint in Leptoceratops also influences the path a powerstroke can take (see below) and is an additional reason for the departure of action in these two taxa. Nevertheless, the basic jaw action of Hungarosaurus traces a similar, albeit, disjointed path.

Apparent from the discussion above is the significant departure of Leptoceratops from all other known models of mastication within Dinosauria. Ceratopsians are entirely lacking in adaptations for cranial kinesis and could not use this type of motion to facilitate mastication. Ceratopsian skulls are tightly sutured, preventing any lateral rotation of the maxillae (Holliday & Witmer, 2008). Furthermore, the quadrate bones have a tongue and grove association with the squamosals and are buttressed posteriorly by the paraoccipital processes of the exoccipital bones, precluding both pleurokinesis and streptostyly (Dodson, 1993). The mandibles are similarly impaired; in Leptoceratops the caudodorsal process of the predentary bone displays a projection that is tightly locked into a pit at the rostrodorsal end of the dentary (Tanoue, You & Dodson, 2010). Furthermore, the caudoventral processes of the predentary overlap the lateral margins of the dentaries, so that no rotational or translational motion could have occurred between these elements. The quadrate-articular jaw joint is thus the only site available to produce the masticatory orbit recorded in the microwear of Leptoceratops.

The semicircular character of dental microwear striations in Leptoceratops resulted from a combination of the aforementioned muscular actions directing motion at the jaw joint. As the chewing cycle began, the quadrate started at the caudal rim of the mandibular glenoid, causing apicodistally oriented striations on dentary teeth. When the quadrate reached the base of the glenoid, microwear on teeth transitioned to a mesiodistal orientation, and then to an apicomesial orientation as further adduction caused the quadrate to ride up the rostral wall of the glenoid. An analogous rostrocaudal masticatory orbit within an akinetic skull occurs in several mammalian groups, including multituberculates (Wall & Krause, 1992), haramiyid allotherians (Butler, 2000), and some rodents (Rose, 2006). Palinal occlusion also occured in cynodonts, but microwear and tooth morphology show no evidence of orbital motion of the mandible (Crompton, 1972; Rybczynski & Reisz, 2001). The presence of mammal-like orbital motion in Leptoceratops is striking because of the structural differences in the jaw joint between dinosaurs and mammals. In mammals, the condyle is on the dentary bone and the glenoid depression is on the underside of the skull in the squamosal. Dinosaurs are essentially the reverse, as they retain the ancestral amniote condition of a condyle on the quadrate of the skull and the glenoid depression in the articular of the lower jaw (Norman & Weishampel, 1991). This retention of the ancestral condition suggests that complex motion of the lower jaw, involving multiple vectors in an orbital motion, need not require a mammalian jaw architecture for its development, and it broadens our understanding of comparative biomechanics.

The greatest biodiversity of ceratopsians occurs during the latter half of the Cretaceous, a time known for the diversification and revolution of terrestrial communities by angiosperm plants (Lloyd et al., 2008). However, recent investigations have questioned or shown no correlation in diversification between dinosaur groups and angiosperms (Weishampel & Jianu, 2000; Barrett & Willis, 2001; Lloyd et al., 2008; Butler et al., 2010). The recognition of a novel chewing adaptation in a ceratopsian dinosaur suggests that previous assignments (Weishampel & Norman, 1989; Weishampel & Jianu, 2000) of taxa to masticatory groupings (orthal pulper, orthal slicer, etc.) may have been too gross. The disparate chewing mechanisms of Euoplocephalus and Hungarosaurus coupled with a depauperate understanding of chewing in other ankylosaurs demonstrates the need for refining the details of jaw action in herbivorous dinosaurs. This refinement, in addition to other areas of inquiry (sampling bias, spatio-temporal diversity and abundance) may bring much needed resolution to the ongoing question of angiosperm-dinosaur coevolution.

The discovery of circumpalinal mastication in a ceratopsian that has been known for over 100 years provokes questions concerning the distribution of this jaw action. Like clinolineal mastication in psittacosaurids (Sereno, Xijin & Lin, 2010), is circumpalinal chewing relatively scarce, limited to Leptoceratops or Leptoceratopsidae, or does it have much wider distribution within Ceratopsia? One of the specimens examined here suggests, at least, a leptoceratopsid distribution. YPM VPPU 018133 was considered by Ostrom (1978) to be Leptoceratops gracilis, but Chinnery (2004) suggested a possible assignment to Prenoceratops based on stratigraphic grounds (the fossil was from the Meeteetse Formation, potentially older than the rocks that yield L. gracilis). Regardless of the assignment of this material, at a minimum it corroborates circumpalinal mastication in Leptoceratops gracilis, and at most hints at a wider distribution of this chewing style within Leptoceratopsidae.

Recent work also addresses the question of wider distribution, and indicates that circumpalinal mastication was not present in derived ceratopsids (Varriale, 2011; Mallon & Anderson, 2014). However, this leaves a large paraphyletic portion of Ceratopsia for which jaw action is unknown, and as such the ancestral action for all ceratopsians is also unknown. Clinolineal mastication could extend to the base of Ceratopsia or be limited to Psittacosauridae. If limited, then circumpalinal or orthopalinal mastication may be the ancestral conditions. A fourth, as-yet undiscovered action may be present there, or the orthal mechanism that seems to be present in the ceratopsian outgroup Pachycephalosauria (Sues & Galton, 1987; Varriale, 2015) and many other ornithischians (Thulborn, 1971; Weishampel, 1984; Crompton & Attridge, 1986; Norman & Weishampel, 1991; Barrett, 1998; Barrett, 2001), could have been retained at the ceratopsian base. To further support the results here for Leptoceratops, and answer the aforementioned questions, a much wider range of taxa will need to be sampled, including other non-ceratopsid neoceratopsians as well as basal ceratopsians outside of Psittacosauridae. This expanded analysis is currently underway by F Varriale.

Recognition of circumpalinal chewing within Leptoceratops adds a distinct style of mastication to the dinosaurian repertoire, one that is fundamentally different from previously understood mechanisms involving cranial or mandibular kinesis. This suggests that some dinosaurs may have possessed a mammalian level of masticatory prowess and biomechanical diversity, achieving this convergence through novel independent adaptations of the masticatory apparatus.

Supplemental Information

Article S1 Visual Basic code used in analysis. Includes instructions for use, output interpretation, code, and modification of code for use by other researchers

Click here for additional data file.

Article S2 Excel spreadsheet containing embedded visual basic macro, and columns of example data for use when reading macro instructions

Click here for additional data file.

Data S1 Excel spreadsheet containing striation orientation data used in calculating summary and inferential statistics

Excel spreadsheet containing striation orientation data (in degrees) used to generate graphs and tabulate summary/inferential statistics of Table 1 and Figs. 1, 2, and 4. Teeth from which these data were collected are identified as column headers. LD, Left dentary, LM, Left maxillary, RD, Right dentary, and RM, Right maxillary. Alveolar position of each tooth is indicated by number.

Click here for additional data file.

Thanks to Andrew Farke, Jordan Mallon, David Weishampel, and Mark Teaford for comments and discussion that improved initial drafts of this manuscript. Paul Barrett and Jeremy Green provided thorough and insightful reviews that enhanced the quality of the narrative. Walter Joyce (YPM), Carl Mehling (AMNH), Natalia Rybczynski (CMN), and Kieran Shephard (CMN) provided access to specimens.

Additional Information and Declarations

Competing Interests

Author Contributions

Data Availability

The author declares there are no competing interests.

Frank J. Varriale conceived and designed the experiments, performed the experiments, analyzed the data, contributed reagents/materials/analysis tools, wrote the paper, prepared figures and/or tables, reviewed drafts of the paper.

The following information was supplied regarding data availability:

The raw data has been supplied as Data S1.

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
