# Peer review of "Dental microwear reveals mammal-like chewing in the neoceratopsian dinosaur Leptoceratops gracilis"

_PeerJ, doi:10.7717/peerj.2132_

## Round 0.1 · original submission · Minor Revisions

Both reviewers did careful, thorough reviews and found much of value in your manuscript. Both also found some areas where the manuscript could be improved. All of their suggestions are pertinent and reasonable, and I encourage you to carefully consider them in revising the manuscript. Congratulations on an interesting and novel finding and a solidly-constructed manuscript. I'll look forward to seeing an even better version soon.

·

Basic reporting

The author has done an excellent job of meeting PeerJ's policies and I have no comments on basic reporting. Great job!

Experimental design

The experimental design is very thorough. The question and hypothesis, as well as methods are quite clear and explicit. The author should be commended for his attention to detail with regards to using Microware 4.02 and teasing out the appropriate data from striation orientation. His revised calculation of the value R is intriguing and his attention to detail with regard to analyzing microwear on each tooth in series is excellent. The standards of PeerJ are met in this regard and it is pleasing to see newer microwear researchers taking the time to appropriately describe the finer details of how they collect and analyze their data.

However, there are three potential issues that need to be addressed: 1) I could not find reference to any mention of pre-analysis taphonomic screening of the tooth surfaces. It is always necessary when working with fossil material (particularly older material) to address any question as to the potential originality of tooth scars - such questions are always raised on fossil microwear analyses, and appropriately so. Such screening procedures are quite simple and are detailed in Teaford (1988: Scanning electron microwear diagnosis of wear patterns versus artifacts on fossil teeth) and King et al. (1999: Effect of taphonomic processes on dental microwear). These papers can be retrieved with an easy scholar google search. Basically, the author should make sure that, following these procedures, there is no question as to their ante-mortem nature of the scratches. Assuming they are indeed original (which I completely think they are, given their consistent orientation on different tooth surfaces, but best to be sure and cover your bases), it is necessary to include a simple sentence or two to this effect in the methods and results. This is a simple edit. 2) I also could not find reference to a mention of a standardized orientation of the specimen relative to the electron beam or detector during analysis. Different angles of tilt relative to the beam and detector during SEM analysis can cause extinction of features that are orientated in specific planes. Thus, if this is not standardized, then it is possible to tilt the specimen to accidentally cause at specific orientations to not be visible on the resulting SEM images. This may or may not be a complication in this analysis - having not actually worked with Leptoceratops teeth or the casts themselves and how they were analyzed, it is difficult to assess this remotely. The author is referred to the following article: 'GALBANY J, MARTÍNEZ LM, PÉREZ-PÉREZ A, 2004: Tooth Replication Techniques, SEM Imaging and Microwear Analysis in Primates: Methodological Obstacles. Anthropologie (Brno) 42, 1: 5-6,' which address some of the potential for extinction of striations at non-standardized specimen orientations. This should be considered in the revision. 3) The manuscript refers to scratches on the "occlusal surface" of teeth, but there is little to no mention of what specific dental issue is being analyzed. Again, I am not intimately familiar with the tooth morphology of this species, but the recent work by Erickson et al. (2015) on Triceratops, which the author cites, shows that different dental tissues (of different hardness) are exposed on the wear facets in that species. Specifically, enamel, mantle dentin, and orthodentin. Are these images just focusing on enamel surfaces, or are dentin wear surfaces also included? I do not think that just because dentin features were or were not analyzed will have any significant bearing on the results, as the focus of this paper is the orientation of striations that are clearly continuous across the tooth surface, regardless of the tissue being analyzed. However, given Erickson's recent paper, I think it would be interesting and worthwhile to mention (and even illustrate on the figures) where and if enamel transitions to dentin and vice versa.

Validity of the findings

There are some issues with the wording of the conclusions that need to be addressed. Although I completely agree that THIS individual of Leptoceratops was chewing in a unique fashion shortly before his/her death, the finding of such a pattern in one skull is hardly grounds for concluding that ALL Leptoceratops chewed in this manner. It is possible (although admittedly not likely) that this is perhaps an aberrant individual, who sustained some injury early in life that caused him/her to chew differently. Or, perhaps something happened shortly before death that forced this individual to chew differently (given the extremely high turnover rate in microwear formation). I think this paper certainly presents a great opportunity for demonstrating that such unique masticatory patterns may have evolved in dinosaurs, but the concession that this study is based on only a single individual absolutely needs to be taken into account. It seems that in the conclusions that author begins to claim victory that all specimens of Leptoceratops and perhaps other ceratopsians achieved this remarkable adaptation. Although this is very possible and it is definitely an exciting possibility, the data are still based on a single instance. Indeed, other ceratopsians have not yet been found to preserve these curvilinear scratches, which means that this conditions currently should be treated as the exception, rather than the norm. Microwear studies have shown that opportunistic feeding events and seasonal changes in diet can lead to statistically significantly differences in quantitative patterns in mammals - why would the same not be present in dinosaurs? I think it would be best to tone down the conclusions a bit and qualify absolute statements a bit more.

For example:
1.) Line 269 should be reworded to say "Recognition of circumpalinal chewing in this individual of Leptoceratops suggests that unique styles of mastication may have evolved in dinosaurs..." (the inclusion of phrases such as "this individual" and "may have" are critical at this preliminary stage).
2). Line 272 should be reworded to say "This suggests that some dinosaurs MAY have possessed a mammalian level of masticatory prowess and biomechanial diversity...future work on ceratopsian microwear will test the hypotheses presented here."

These changes do not hamper or call into question these possibilities the author raises, but it acknowledges that this still could be a single instance and the data are limited. Thus, more work is needed - it would be good for the author to expound on what other specimens or groups need to be studied to further support these findings. Make some predictions, present some new hypotheses, but acknowledge there is not a repeatable pattern among different individuals observed YET. It is always better when working with microwear data, which is subject to taphonomic changes and high rates of turnover, to tread with caution in light of data from a single individual.

Additional comments

On the whole, great job! This is fascinating work and I can't wait to see what comes of this type of research in the future!

·

Basic reporting

The MS reports a new jaw mechanism that is currently unique to the ceratopsian dinosaur Leptoceratops. This jaw mechanism has not been proposed previously and adds to the diversity of known feeding strategies in Dinosauria. The article is clearly written and easy to follow and as far as I can tell conforms to PeerJ's policies and formatting requirements. Figures and their accompanying legends adequately convey the information presented within the text and help with the interpretation of the suggested function. The work presented is self-contained and coherent: it does not rely on other work to be published elsewhere. All of the raw data necessary to replicate the work has been included in the supplement and I commend the author on documenting his analytical procedures with clarity and in detail. My one major criticism would be that the author has not cited all of the relevant literature in this area (particularly with respect to papers published in the last five years) and this should be updated and the relevant sections of the manuscript expanded very slightly to take this more recent information into account.

Experimental design

This is original primary research that has been prepared for publication from an unpublished PhD thesis written by the author (I was previously aware of the thesis). The hypothesis to be tested is clearly stated and the criteria for passing/failing the tests of this hypothesis are listed. The results presented and the interpretations built thereon are novel and have not been published previously by this author or by others. The data collection proceedures are described to a high level of detail, making it easy for others to replicate this work. Similarly, the author has gone to some lengths to make his analytical proceedures very clear. All of the methods used are appropriate for the questions asked and seem to have been applied well. There are no ethical concerns in this instance.

Validity of the findings

The data collected are novel and the analyses performed on scratch orientation and direction are robust. Results are clearly presented and consistent with the authors preferred interpretation relating to directions of jaw movement during the chewing cycle. There is little speculation here and the related anatomical discussions over jaw musculature etc. also appear to be robust.

Additional comments

I've very few general comments to offer as in general the MS is self-explanatory and clearly expressed. My one major concern relates to the authors apparent lack of familiarity with the recent literature and the need to bring this into the introduction and discussion to establish a proper context for his results and discussion (and to give due credit to others). These can be summarised thus:

Line 29, rather than cherry pick one or two dinosaur studies here I would mention the most comprehensive reviews of jaw function across the group and a larger number of studies on other non-mammalian amniotes (including non-mammalian synapsids). If attempting to be comprehensive here there are many other recent dinosaur studies that should be listed including Norman et al. (2012) and Sereno (2012) [on heterodontosaurids], Tanoue et al. (2009), Sereno et al. (2010) and Erickson et al. (2015) [on other ceratopsians], Osi et al. (2014) [on ankylosaurs], and Williams et al. (2009), Cuthbertson et al. (2012) and Erickson et al. (2012) [hadrosaurs].

Lines 31, 34, 37, 40. Dodson et al. (2004), Horner et al. (2004) and Dodson (1996) are not studies of feeding in these groups, but general reviews. These papers should not be cited in this context and instead the specialist papers describing feeding should be listed instead, including for example Ostrom (1961, 1966), Tanoue et al. (2009), Williams et al. (2009), Cuthbertson et al. (2012) and Erickson et al. (2012, 2015). You list some of these papers, but not always as often as they should be.

Line 206: Palinal components in dinosaur jaw mechanisms have been proposed on numerous previous occassions, but are not acknowledged. They have been mentioned in diplodocid sauropods (Barrett & Upchurch 1994), hadrosaurs (Ostrom 1961, Williams et al. 2009), ceratopsids (Sampson 1996, Barrett 1998 - both PhD dissertations) and Hypsilophodon (Williams 2010 - downloadable PhD dissertation).

Line 219: This issue is explicitly addressed in several recent papers that should be cited and have their results mentioned here (Williams et al. 2009; Cuthbertson et al. 2012; Erickson et al. 2012).

Line 224: Other authors have also commented on heterodontosaurid jaw action. These other papers include Crompton & Attridge (1986), Sereno (2012) & Norman et al. (2012). Their conclusions should also be mentioned in this context.

Line 229: Please note this type of jaw mechanism has only been confirmed for Euoplocephalus thus far and has not been tested in other ankylosaurids. Also, this is not the only ankylosaur for which complex jaw mechanisms have been proposed - see also Osi et al. (2014) on the nodosaurid Hungarosaurus - again these results should also be mentioned in this context.

Line 266: Would be good to cite Lloyd et al. (2008) in the context of the mid-Cretaceous terrestrial revolution.

Line 267: The same conclusions were reached also by Barrett & Willis (2001).

Other minor comments:

Line 195: pterygoideus is misspelt.
Line 201: Wouldn't the m. adductor mandibulae superficialis also have a minor posterior component that would assist the m. adductor posterior in the caudodorsal movement of the jaw?

---

## Round 0.2 · accepted · Accept

Thank you for your thoroughness in addressing the reviewers' concerns. I am happy to accept the revised manuscript for publication in PeerJ.

The decision of whether or not to publish the peer reviews alongside the paper is entirely yours, and will not affect how your paper is handled going forward. However, I encourage you to do so. This is a great example of strong manuscript being made even stronger by a constructive review process. More importantly, both reviewers chose to sign their reviews, and making the reviews public allows the reviewers to receive more credit for their efforts, and also contributes to the emerging culture of fairness and transparency in editing and peer review.